# Synthesis of Gold-Platinum Core-Shell Nanoparticles Assembled on a Silica Template and Their Peroxidase Nanozyme Properties

**DOI:** 10.3390/ijms23126424

**Published:** 2022-06-08

**Authors:** Xuan-Hung Pham, Van-Khue Tran, Eunil Hahm, Yoon-Hee Kim, Jaehi Kim, Wooyeon Kim, Bong-Hyun Jun

**Affiliations:** 1Department of Bioscience and Biotechnology, Konkuk University, Seoul 05029, Korea; phamricky@gmail.com (X.-H.P.); greenice@konkuk.ac.kr (E.H.); hilite2201@naver.com (Y.-H.K.); susia45@gmail.com (J.K.); buzinga5842@konkuk.ac.kr (W.K.); 2VNUK Institute for Research and Executive Education, The University of Danang, Danang 550 000, Vietnam; khue.tran@vnuk.edu.vn

**Keywords:** gold-platinum bimetallic nanoparticles, nanoparticle, gold-platinum-assembled silica nanostructures, nanozyme, peroxidase-like activity

## Abstract

Bimetallic nanoparticles are important materials for synthesizing multifunctional nanozymes. A technique for preparing gold-platinum nanoparticles (NPs) on a silica core template (SiO_2_@Au@Pt) using seed-mediated growth is reported in this study. The SiO_2_@Au@Pt exhibits peroxidase-like nanozyme activity has several advantages over gold assembled silica core templates (SiO_2_@Au@Au), such as stability and catalytic performance. The maximum reaction velocity (V_max_) and the Michaelis–Menten constants (K_m_) were and 2.1 × 10^−10^ M^−1^∙s^−1^ and 417 µM, respectively. Factors affecting the peroxidase activity, including the quantity of NPs, solution pH, reaction time, and concentration of tetramethyl benzidine, are also investigated in this study. The optimization of SiO_2_@Au@Pt NPs for H_2_O_2_ detection obtained in 0.5 mM TMB; using 5 µg SiO_2_@Au@Pt, at pH 4.0 for 15 min incubation. H_2_O_2_ can be detected in the dynamic liner range of 1.0 to 100 mM with the detection limit of 1.0 mM. This study presents a novel method for controlling the properties of bimetallic NPs assembled on a silica template and increases the understanding of the activity and potential applications of highly efficient multifunctional NP-based nanozymes.

## 1. Introduction

Nanozymes, a new functional nanomaterial with enzyme-like catalytic activity, have several advantages when compared with natural enzymes, including high stability in harsh environments, low production costs, large specific surface areas, and customizable catalytic activities based on size, morphology, and composition [1,2,3,4,5,6,7,8,9,10,11]. A series of nanomaterials made from metals, metal oxides, and other materials including Pt [12,13,14], Au [15,16,17,18], Ag [19], Cu [20], Fe_3_O_4_ [21], CeO_2_ [22], MnO_2_ [23], Mn_3_O_4_ [24,25], conducting polymers [26], metal–organic frameworks [27], carbon nanomaterials [28], and single-atom catalysts [29] have been prepared for use as nanozymes. These nanozymes have been used as bio(chemical) sensors, in immunoassays, for drug delivery, and as antibacterial agents [3,12,13,30,31,32,33,34]. While the physical and chemical properties of nanoparticles (NPs) can be adjusted by changing their size, shape, or structure, monometallic NPs are limited by other physical or chemical properties such as size-dependent optical properties, electronic properties, and thermal and catalytic effects that control the chemical compositions of NPs [35].

Several studies have been conducted on bimetallic NPs owing to the increased awareness of the options to adjust their physical and chemical properties. Compared with monometallic NPs, bimetallic NPs allow a greater adjustment of their magnetic, optical, and catalytic properties via adjustments of their composition and chemical configuration [36]. Bimetallic NPs have unique structure- and composition-dependent properties that allow the optimization of enzyme-like activities more than those of single-metal NPs [37,38]. The Pt family of bimetallic NPs has been developed using various morphologies including core–shell structures, solid solution alloys, intermetallic alloys, and phase-segregated structures. Owing to the synergistic effect caused by the presence of Pt NPs, a super catalyst for electrochemical reactions, which are designed to maximize Pt utilization, tune the energetics, and assemble the geometry of exposed Pt atoms for high catalytic reactivity and selectivity, has been created [39,40]. Bimetallic NPs exhibit various enzyme-like activities including peroxidase, catalase, polyphenol oxidase, ferroxidase, and superoxide dismutase activities [40,41,42,43,44]. Au@Pt bimetallic NPs have intrinsic peroxidase-like activities [45,46,47], have been studied extensively for electrochemical formic acid and alcohol oxidation, and Pt-rich compositions exhibit enhanced electrocatalytic activity [48,49]. Au@Pt NPs have extensive catalytic properties as they can effectively scavenge superoxide free radicals, H_2_O_2_, or formic acid, enabling their use in fuel cells, hydrogenation, air purification, anti-aging therapy, and cancer therapy [48,50,51,52]. However, Pt tends to aggregate in catalytic reactions, resulting in reduced catalytic activity [53].

Au@Ag or Au NPs immobilized on SiO_2_ NPs have been developed recently [54,55,56,57,58]. In these nanostructures, the SiO_2_ NPs serve as a template and the metal NPs are introduced to the surface using the seed-mediated growth method. Because of the SiO_2_ core, the metal NPs assembled on SiO_2_ nanostructures are cost-effective, more stable during surface modification, easily separatable from the reaction solution, aggregate less than fine metal NPs, and redisperse into solution with enhanced separation compared with single metal NPs. Furthermore, the effects of various synthesis factors on metal NPs assembled on SiO_2_ have been studied using near-infrared surface enhanced Raman scattering nanoprobes for bioimaging [55,56]. However, research regarding the high peroxidase-like activity of the metal NPs assembled on SiO_2_ is limited. Therefore, the development of an improved method for preparing metal NPs assembled on SiO_2_ templates is needed to enhance peroxidase activity. This study improves the understanding of the activity and applications of highly efficient multifunctional nanozymes.

In this study, gold-platinum NPs are prepared on a silica core template (SiO_2_@Au@Pt) using seed-mediated growth combined with the dropping method and utilize it as a nanomaterial to detect hydrogen peroxide (H_2_O_2_) as a model because H_2_O_2_ plays an important role in cellular metabolism and in various industry such as food industry, gas sensors, pharmaceuticals, catalysis, environment, and solar energy [59,60,61]. Therefore, the need to develop a low cost, simple, fast, and sensitive method for monitoring the concentration of H_2_O_2_ is of practical significance in both industry and academia. The preparation of SiO_2_@Au@Pt consists of two steps: tiny Au seeds are embedded on the SiO_2_ surface, and the addition of a Pt^2+^ precursor in the presence of ascorbic acid (AA) reductant allows the deposition and growth of the Pt layer. First, a suspension of Au seeds (approximately 2.5 nm) was prepared using tetrakis(hydroxymethyl)phosphonium chloride (THPC) and HAuCl_4_. Then, the Au seeds were mixed with aminated SiO_2_ NPs (approximately 160 nm) overnight to obtain SiO_2_@Au seeds, as previously reported [54,55,57,58,62,63,64]. On the SiO_2_@Au seeds, the reduction of Pt^2+^ to Pt(0) was directly induced using AA, a mild reducing agent, in the presence of polyvinylpyrrolidone (PVP). These mildly reducing conditions allow a greater control over the growth of the Pt layer as the reaction proceeds much slower than that in strongly reducing conditions [65]. A low concentration of the Pt^2+^ precursor and AA were added onto the SiO_2_@Au seeds in 5 min intervals to enable the precise control of the size of the Pt NPs. After the optimization of synthesis, SiO_2_@Au@Pt was used as a peroxidase-like nanomaterial to detect H_2_O_2_ efficiently.

## 2. Results and Discussion

### 2.1. Preparation of Au@Pt NPs-Assembled Silica Nanostructures

The seed-mediated growth from SiO_2_@Au seeds was used to prepare the gold-platinum-embedded silica nanospheres (SiO_2_@Au@Pt). First, the surface SiO_2_ template was assembled using small Au NPs (2.6 ± 0.52 nm) (Figure 1a). Then, 10 mM H_2_PtCl_4_ and AA solutions were added dropwise into the dispersion of SiO_2_@Au seeds to obtain a final Pt^2+^ concentration of 200 µM. The Au@Pt NPs on the surface of the SiO_2_ core were bigger (3.6 ± 0.56 nm) than SiO_2_@Au seeds. The presence of Pt on the surface of SiO_2_@Au was confirmed by the line energy dispersive X-ray (EDS) mapping of SiO_2_@Au@Pt (Figure 1b and Appendix A). In Figure 1b, the signal of both Pt and Au elements could obtain in the image. The quantitative EDS analysis in Appendix A of SiO_2_@Au@Pt synthesized at 200 µM Pt^2+^ consists of 74.6% Pt and 23.6% Au as mentioned in Appendix A.

The UV-Vis spectroscopy of SiO_2_@Au@Pt was conducted when the Pt^2+^ solution was added into the SiO_2_@Au seed suspension in the presence of AA. The absorbance intensity of SiO_2_@Au showed a slight increase at ~500 nm. This peak was suppressed in the absorbance spectrum of SiO_2_@Au@Pt but the spectrum of SiO_2_@Au@Pt increased in the UV region (Figure 1c) owing to the Pt layer, consistent with previous results of pure Pt hydrosol [66,67]. The results of TEM, EDS mapping, UV-Vis spectrum of SiO_2_@Au@Pt indicated that Pt was deposited to the SiO_2_@Au.

### 2.2. Peroxidase-like Activity of SiO_2_@Au@Pt NPs

The peroxidase-like activity of SiO_2_@Au@Pt NPs was evaluated using the oxidation reaction of 3,3′,5,5′-tetramethylbenzidine (TMB) substrate prepared in a buffer (pH = 4) containing TMB or a TMB-H_2_O_2_ mixture. TMB oxidation involves the exchange of two electrons. When TMB transfers one electron to form TMB^+^, the solution changes from colorless to blue. However, TMB^+^ is unstable in acidic conditions and must oxidize to TMB^2+^, forming a yellow solution [68]. In this study, the TMB, H_2_O_2_, and TMB+H_2_O_2_ solutions without SiO_2_@Au@Pt NPs were colorless and no absorbance peaks were observed at 453 nm (Figure 1d), indicating that there was no peroxidase-like catalytic activity in the absence of SiO_2_@Au@Pt. A small absorbance band was observed for the SiO_2_@Au@Pt + TMB solution, and a strong absorbance band with peaks at 370 and 652 nm was observed for the SiO_2_@Au@Pt + TMB-H_2_O_2_ solution. The SiO_2_@Au@Pt + TMB-H_2_O_2_ solution changed from blue to yellow and an absorbance band with a peak at 453 nm was observed (Appendix A). These results indicate that the conversion of TMB to TMB^2+^ was catalyzed by the SiO_2_@Au@Pt NPs with H_2_O_2_, suggesting that SiO_2_@Au@Pt NPs has a peroxidase-like activity. To confirm the synergic qualities of SiO_2_@Au@Pt, an Au layer was deposited on the surface of the SiO_2_@Au seeds using a 200 µM Au^3+^ solution. The peroxidase-like catalytic activity of SiO_2_@Au@Pt (5 µg) and SiO_2_@Au@Au (5 µg) in TMB and TMB-H_2_O_2_ solutions was then verified (Figure 2a and Appendix A).

The blue color of the SiO_2_@Au@Pt in TMB-H_2_O_2_ is darker and the absorbance intensity at 370 and 652 nm is stronger than those of the SiO_2_@Au@Au suspension. The absorbance intensities of the SiO_2_@Au@Pt suspension were 5.7- and 7.7-fold of those of SiO_2_@Au@Au at 370 and 652 nm, respectively. Similarly, the absorbance intensity of the SiO_2_@Au@Pt suspension at 453 nm was stronger than that of SiO_2_@Au@Au (Appendix A). In addition, the recycling of both SiO_2_@Au@Au and SiO_2_@Au@Pt in the TMB-H_2_O_2_ solution at 453 nm was compared when the NPs were reused five times. The absorbance intensity of SiO_2_@Au@Au in a TMB-H_2_O_2_ solution at 453 nm decreased to approximately 40% while that of SiO_2_@Au@Pt in a TMB-H_2_O_2_ solution at 453 nm decreased to 80%. These results indicate that the catalytic ability of SiO_2_@Au@Pt is better than that of SiO_2_@Au@Au, and that SiO_2_@Au@Pt NPs are reusable and separatable from the reaction mixture.

The catalytic performance of SiO_2_@Au@Pt NPs at TMB concentrations of 0–600 µM was also investigated as the absorbance intensity at 652 nm in TMB-H_2_O_2_ every 3 min (Figure 2b). The absorbance intensity of the oxidation of TMB increased as the TMB concentration increased, following Michaelis–Menten behavior. The signal of SiO_2_@Au@Pt NPs increased as the TMB concentration increased from 100 to 500 µM and reached saturation at 600 µM. The relationship of the TMB concentration and absorbance intensity at 652 nm after incubation for 180 s was plotted according to the Lineweaver–Burk equation to calculate the maximum reaction velocity (V_max_) and the Michaelis–Menten constants (K_m_) (Figure 2d). The kinetic activity of SiO_2_@Au@Pt at various TMB concentrations (100–500 µM) revealed a linear relationship. The K_m_ was 417 µM and the V_max_ was 2.1 × 10^−10^ M^−1^∙s^−1^. The K_m_ of SiO_2_@Au@Pt indicates a higher affinity of SiO_2_@Au@Pt for TMB compared with that for horseradish peroxidase enzyme (K_m_ = 438 µM). The K_m_ of SiO_2_@Au@Pt was higher than those of Au NPs (K_m_ = 123 µM), SiO_2_@Au@Au NPs (Km = 60 µM), glucose oxidase-conjugated Au-attached SiO_2_ microspheres (K_m_ = 208 µM), MnO_2_ NPs (K_m_ = 83 µM), and latex-conjugated MnO_2_ NPs (K_m_ = 99 µM). The K_m_ of SiO_2_@Au@Pt was lower than those of Au NPs-decorated porous silica microspheres (K_m_ = 523 µM) and Prussian-blue-decorated latex NPs (K_m_ = 2.19 mM) [55]. The V_max_ of SiO_2_@Au@Pt was higher than that of SiO_2_@Au@Au (V_max_ = 2.1 × 10^−10^ M^−1^∙s^−1^), indicating that SiO_2_@Au@Pt oxidizes TMB at a faster rate than SiO_2_@Au@Au. SiO_2_@Au@Pt had a comparatively more stable catalytic activity that remained at 80% after five reuses (Figure 2b).

### 2.3. Effects of Synthesis and Experimental Conditions on the Catalytic Activity of SiO_2_@Au@Pt NPs

Various concentrations of the Pt^2+^ precursor (100–400 μM) were added to the SiO_2_@Au seeds. The size of Pt increased as the Pt^2+^ concentration increased (Figure 3a and Appendix A). In particularly, the size of Au@Pt synthesized at 100, 200, 300, 400 µM Pt^2+^ were 3.1 ± 0.61; 3.6 ± 0.56; 3.9 ± 0.59; 4.6 ± 0.62 nm, respectively. At a high concentration of Pt^2+^ (>200 µM), the Au@Pt on the SiO_2_ surface partly merged. The quantitative EDS analysis of Pt and Au elements on the surface of SiO_2_ synthesized at various Pt^2+^ concentration was carried out to investigate the composition of the Au@Pt on the SiO_2_ surface and the results were shown in Appendix A. All Au@Pt NPs contained both Pt and Au elements, but their Au and Pt components were different. The atomic Pt component increased from 71.59% to 87.07% while the atomic Au component decreased from 28.41 to 12.93% when Pt^2+^ increased from 100 µM to 400 µM. Therefore, the reciprocal of Pt and Au increased from 2.5 (at 100 µM Pt^2+^) to 6.7 (at 400 µM Pt^2+^). The results matched to the TEM images, and it indicated that Pt was gradually deposited on the surface of SiO_2_@Au.

The absorption spectra of SiO_2_@Au@Pt were collected (Figure 3b). The absorbance intensity at 300–800 nm increased as the concentration of the Pt^2+^ precursor increased, indicating the formation of larger Au@Pt NPs (Appendix A) that subsequently affected the proximate interparticle distance. The absorbance of the suspension broadened as the size of the Au@Pt NPs increased (Appendix A). The growth of Au@Pt NPs on SiO_2_ could be controlled well (Figure 3).

The correlation between the peroxidase-like activity of SiO_2_@Au@Pt NPs and the concentration of Pt^2+^ was investigated (Figure 3c). The peroxidase-like activity of SiO_2_@Au@Pt NPs synthesized with 100–400 μM Pt^2+^ was estimated using a TMB assay. The SiO_2_@Au seeds showed very weak peroxidase-like activity because of the lack of spaces on the SiO_2_@Au NPs and the small Au NPs on the SiO_2_ template, resulting an insufficient surface area for the catalytic reaction between the Au NPs and the TMB-H_2_O_2_ mixture [55]. The UV-Vis absorption spectra of all SiO_2_@Au@Pt NPs showed an absorbance peak at 453 nm (Figure 3c–e), indicating that all of the SiO_2_@Au@Pt NPs had peroxidase-like activity that was dependent on the initial Pt^2+^ concentration.

In contrast, SiO_2_@Au@Pt NPs treated with a Pt^3+^ precursor concentration >100 µM had high peroxidase-like activity. As the concentration of Pt^2+^ increased, the size of the Au@Pt NPs on the SiO_2_@Au@Pt NPs increased from 3.1 to 4.6 nm, which increased the surface area for reactions between NPs and reactants. Therefore, the catalytic activity of SiO_2_@Au@Pt NPs increased as the concentration of the Pt^2+^ precursor increased from 100 to 200 μM because of the formation of sublayers or a monolayer of Pt on the surface of SiO_2_@Pt [45,67]. This result consistent with the previous report where the increase of Pt component of Au-Pt led an better catalytic activity because the alloying of Pt with Au can change the electronic structure of Pt, leading the catalytic performance of Au@Pt changes [11,37,69]. Although the Au@Pt NPs grew as the concentration of the Pt^2+^ precursor increased, the peroxidase reaction of SiO_2_@Au@Pt treated with >200 µM of the Pt^2+^ precursor did not increase because TMB and H_2_O_2_ cannot gain access to inner part of the thick Pt layer [66]. It means that the catalytic activities of SiO_2_@Au@Pt reached the highest value at the Pt/Au ratio of 3.2 and decreased when Pt^2+^ concentration increased further. Therefore, 200 µM of Pt^2+^ precursor-treated SiO_2_@Au@Pt NPs, which exhibited high peroxidase-like activity, were used in subsequent experiments.

Reaction conditions affect the catalytic activity of nanozymes similarly to the effects of reaction conditions on enzymes [21,70,71,72,73,74]. Therefore, the peroxidase-like activity of different amounts of SiO_2_@Au@Pt NPs were investigated at different incubation times, pH values of the buffer solution, and TMB concentrations (Figure 4).

The amount of SiO_2_@Au@Pt varied from 0.02 to 20 μg (Figure 4a and Appendix A). The absorbance intensity of SiO_2_@Au@Pt in a TMB-H_2_O_2_ solution at 453 nm increased as the amount of SiO_2_@Au@Pt increased from 0.02 to 5 μg. When the amount of SiO_2_@Au@Pt increased to >10 μg, the poor solubility of TMB in the aqueous solution resulted in significant aggregation, inducing precipitation with oxidation [55].

The absorbance intensity increased as the incubation time increased from 5 to 15 min (Figure 4b and Appendix A). The absorbance intensity reached saturation after 15 min of incubation.

The highest peroxidase catalytic activity was obtained at a pH of 4.0, at which TMB dissolved maximally and H_2_O_2_ was the most stable (Figure 4c and Appendix A), which is consistent with the results of previous studies [17,21,75,76,77,78].

The absorbance of TMB^2+^ at 453 nm increased as the TMB concentration increased and reached the saturation at 600 μM TMB (Figure 4d and Appendix A).

### 2.4. Effects of H_2_O_2_ Concentration on Peroxidase-like Activity of SiO_2_@Au@Pt NPs

After optimizing the detection conditions, the absorbance intensity was measured at 0–400 nm for the SiO_2_@Au@Pt in 500 μM TMB with various concentrations of H_2_O_2_ (Figure 5). The yellow color of the 5 μg SiO_2_@Au@Pt suspension became darker as the H_2_O_2_ concentration increased to 200 mM, indicating that more TMB^2+^ was produced as the H_2_O_2_ concentration increased. The catalytic activity of SiO_2_@Au@Pt increased as the H_2_O_2_ concentration increased to 200 mM and reached saturation at 300 mM because of aggregation (Appendix A).

A linear curve-fitting procedure was used to calibrate the reaction (Figure 5c). A significant relationship was found between the absorbance intensity at 453 nm and the H_2_O_2_ concentration from 1.0 to 100 mM (calibration curve: y = 0.0185 x + 0.63285, where x is the H_2_O_2_ concentration, y is the absorbance intensity at 453 nm, and R^2^ = 0.99). The theoretical LOD was 1.0 mM, estimated using the 3sblank criterion. This LOD is higher than the LOD of silver NPs modified cellulose nanowhiskers [79], polyoxometalate [80], Ce_2_(WO_4_)_3_, papain [81], Ag-nanoparticle-decorated silica microspheres [82] and magnetic mesoporous silica nanoparticles [83]. These results suggest that this material can be used to detect H_2_O_2_.

## 3. Materials and Methods

### 3.1. Chemicals and Reagents

Tetraethylorthosilicate (TEOS), chloroauric acid (HAuCl_4_), THPC, chloroplatinic acid (H_2_PtCl_6_), 3-aminopropyltriethoxysilane (APTS), AA, TMB, and PVP (MW 40,000) were purchased from Sigma-Aldrich (St. Louis, MO, USA). Ammonium hydroxide (NH_4_OH, 27%), sodium hydroxide (NaOH), ethyl alcohol (EtOH, 99.9%), and sulfuric acid (H_2_SO_4_) were obtained from Samchun (Seoul, Korea). H_2_O_2_ was purchased from Daejung (Siheung, Gyeonggi-do, Korea). Phosphate buffer saline containing 0.1% Tween 20 (PBST, pH 7.4) was obtained from Dynebio (Seongnam, Gyeonggi-do, Korea).

### 3.2. Characterization

The TEM images of the samples were obtained using a JEM-F200 electron microscope (JEOL, Akishima, Tokyo, Japan) at an accelerated voltage of 200 kV. The UV-Vis spectra of the samples were recorded using an Optizen POP UV/Vis spectrometer (Mecasys, Seoul, Korea). The samples were centrifuged using a 1730R microcentrifuge (LaboGene, Lyngen, Denmark).

### 3.3. Synthesis of Gold-Platinum Nanoparticles Assembled on a SiO_2_ Nanostructure (SiO_2_@Au@Pt NPs)

The SiO_2_@Au seed NPs were synthesized as previously reported [54]. Briefly, colloidal Au NPs were prepared from HAuCl_4_ and THPC. Silica templates (approximately 160 nm) were prepared using the modified Stöber method [84]. The surfaces of 50 mg SiO_2_ NPs were modified using amino groups via incubation with 62 μL APTS overnight at 25°C. Animated SiO_2_ NPs (2 mg) were incubated with 10 mL colloidal Au (approximately 2.5 nm) for 12 h at 25°C. After the suspension was centrifuged for 10 min at 8500 rpm and washed with EtOH, 2 mg of SiO_2_@Au seed NPs were dispersed in 2 mL of 1 mg/mL PVP solution. Subsequently, 200 μL of SiO_2_@Au seed (1 mg/mL) suspension was mixed with 9.8 mL of PVP solution. Under stirring, 20 μL of 10 mM HPtCl_6_ solution (in water, Pt^2+^ precursor) and 40 μL of AA reducing agent (10 mM AA in water) were added to the mixture. The mixture was reacted for 5 min under stirring to convert Pt^2+^ to Pt(0). The same volumes of Pt^2+^ precursor and AA were added every 5 min to obtain the desired concentration of Pt^2+^. The SiO_2_@Au@Pt NPs were then carefully washed with EtOH several times using centrifugation at 8500 rpm for 10 min. The washed SiO_2_@Au@Pt NPs were redispersed in 0.1% PBST solution (1 mL) to obtain a 0.2 mg/mL SiO_2_@Au@Pt NP suspension.

### 3.4. Peroxidase-like Activity of SiO_2_@Au@Pt NPs

To verify the peroxidase-like catalytic activity of SiO_2_@Au@Pt NPs, TMB solution (6 mM in EtOH, 100 μL), the SiO_2_@Au@Pt NPs suspension (100 μL) synthesized with 100, 200, 300, and 400 μM Pt^2+^, and freshly prepared H_2_O_2_ solution (2 M in pH 4 buffer, 100 μL) were added to 700 μL of buffer (pH = 4.0). Then, the mixture was incubated at 25°C for 15 min. To terminate the reaction, 1 M H_2_SO_4_ solution (500 μL) was added to each mixture and the resulting mixture was incubated at 25°C for 10 min. The absorbances of the suspensions were measured at 300–1,000 nm using a UV-Vis spectrometer.

### 3.5. Peroxidase-like Activity of SiO_2_@Au@Pt in Various Reaction Conditions

#### 3.5.1. Amount of SiO_2_@Au@Pt NPs

A mixture of 6 mM TMB solution in EtOH (100 μL), 2 M H_2_O_2_ (100 μL), and 700 μL buffer (pH = 4) was added to 100 μL PBST containing 0, 0.2, 2, 5, 10, or 20 μg of SiO_2_@Au@Pt. The resulting solution was incubated at 25 °C for 15 min, then terminated with 1 M H_2_SO_4_ (500 μL). The absorbances were measured at 453 nm using a UV-Vis spectrometer.

#### 3.5.2. Reaction Time

A mixture of a 6 mM TMB (100 μL), a SiO_2_@Au@Pt NP suspension (0.05 mg/mL, 100 μL), and 2 M H_2_O_2_ (100 μL) was added to 700 μL buffer (pH = 4) then incubated at 25°C for various incubation times. The mixture was terminated using 1 M H_2_SO_4_ (500 μM). The absorbances of the mixtures were measured at 453 nm using a UV-Vis spectrometer.

#### 3.5.3. PH Value of the Buffer

A mixture containing 6 mM TMB (100 μL), a SiO_2_@Au@Pt NP suspension (0.05 mg/mL, 100 μL), and 2 M H_2_O_2_ (100 μL) was added to 700 μL of buffer at a pH range from 3.0 to 11.0. The mixture was incubated for 15 min and terminated using 1 M H_2_SO_4_ (500 μL). The absorbances of the mixtures were measured at 453 nm using a UV-Vis spectrometer.

#### 3.5.4. TMB Concentration

A mixture containing 700 μM buffer (pH 4.0), 2 M H_2_O_2_ (100 μL), and 0.05 mg/mL SiO_2_@Au@Pt suspension (100 μL) was added to various concentrations of TMB (1, 2, 3, 4, 5, and 6 mM). The final TMB concentration in the reaction mixture was 0.1, 0.2, 0.3, 0.4, 0.5, and 0.6 mM. After incubating the mixture for 15 min, the reaction was terminated using 1 M H_2_SO_4_ (500 μL). The absorbances of the mixtures were measured at 453 nm using a UV-Vis spectrometer.

## 4. Conclusions

In summary, SiO_2_@Au@Pt NPs were successfully synthesized using the seed-mediated growth method under mild conditions. Compared with SiO_2_@Au@Au, the SiO_2_@Au@Pt NPs exhibited more synergic and stable catalytic abilities that remained at 80% after five uses of 5 μg SiO_2_@Au@Pt NPs. In addition, the peroxidase-like activity of SiO_2_@Au@Pt NPs under various conditions such as the amount of SiO_2_@Au@Pt NPs, pH of the buffer solution, incubation time, and TMB concentration were also investigated, revealing optimized conditions of 5 μg SiO_2_@Au@Pt at pH 4.0, with 15 min of incubation in the presence of 500 μM TMB. SiO_2_@Au@Pt was used to detect H_2_O_2_. The dynamic linear range was obtained from 1 to 100 mM, with an LOD of 1.0 mM. Therefore, this study suggests novel uses of bimetallic metal-assembled silica nanostructures in various fields and provides a suitable method for the development of nanoparticle-based multi-functional nanozymes.

## Figures and Tables

**Figure 1 ijms-23-06424-f001:**
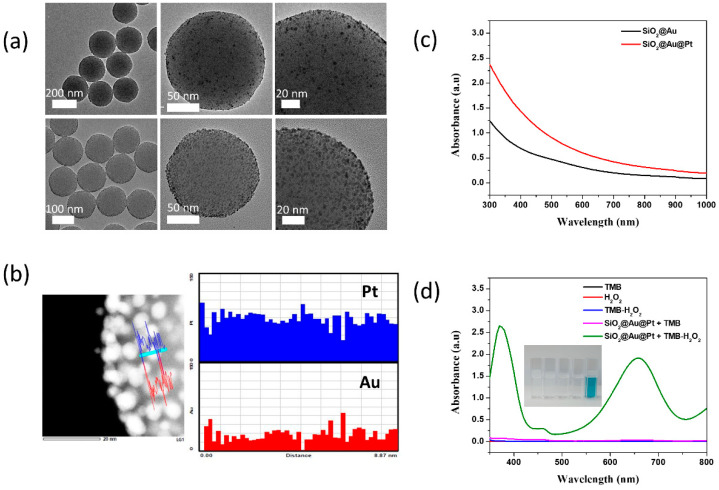
(**a**) Transmission electronic microscopy images of SiO_2_@Au seed and gold-platinum-embedded silica nanospheres (SiO_2_@Au@Pt) at different magnifications. (**b**) Line energy dispersive X-ray mapping of SiO_2_@Au@Pt for Pt and Au elements. (**c**) UV-Vis absorbance spectroscopy of SiO_2_@Au seeds and SiO_2_@Au@Pt NPs. (**d**) UV-Vis absorbance spectroscopy of SiO_2_@Au@Pt in various solutions. The inset shows the colors of the solutions.

**Figure 2 ijms-23-06424-f002:**
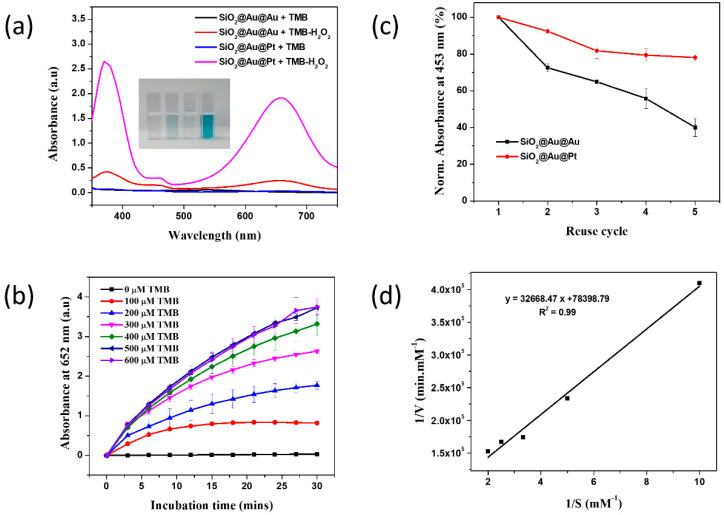
(**a**) UV-Vis absorbance spectroscopy of SiO_2_@Au@Pt and SiO_2_@Au@Au (5 µg) in TMB and in TMB-H_2_O_2_ solutions. The inset shows the colors of the solutions. (**b**) The absorbance of recycled SiO_2_@Au@Au and SiO_2_@Au@Pt is shown in a TMB-H_2_O_2_ solution. (**c**) UV-Vis absorbance spectroscopy and (**d**) Lineweaver-Burk plot at 652 nm for 5 μg SiO_2_@Au@Pt in a TMB-H_2_O_2_ solution (0–600 µM TMB).

**Figure 3 ijms-23-06424-f003:**
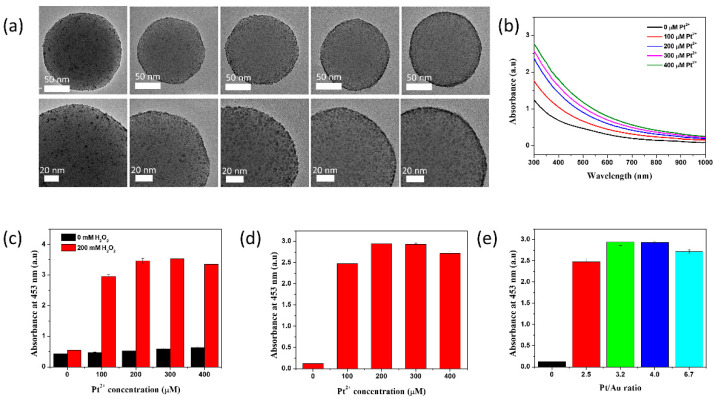
(**a**) TEM images are shown at different magnifications. (**b**) UV–Vis absorbance spectroscopy images of SiO_2_@Au@Pt NPs fabricated with various concentrations of Pt^2+^. (**c**,**d**) Absorbance plots of SiO_2_@Au@Pt NPs and (**e**) Effect of Pt/Au ratio on the absorbance at 453 nm in the presence of TMB and H_2_O_2_ fabricated with various concentrations of Pt^2+^.

**Figure 4 ijms-23-06424-f004:**
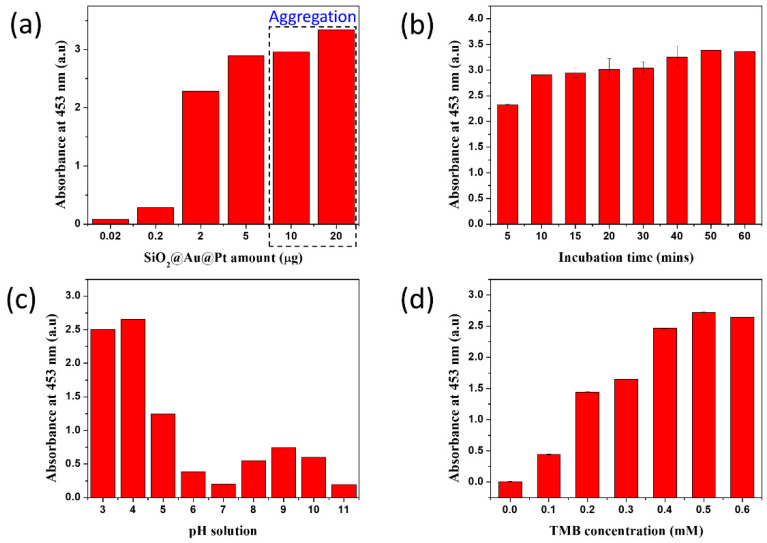
The effects of different conditions on the peroxidase-like catalytic activity of SiO_2_@Au@Pt NPs in a mixture of TMB and H_2_O_2_. (**a**) The amount of SiO_2_@Au@Pt, (**b**) incubation time, (**c**) pH of the solution, and (**d**) TMB concentration were varied.

**Figure 5 ijms-23-06424-f005:**
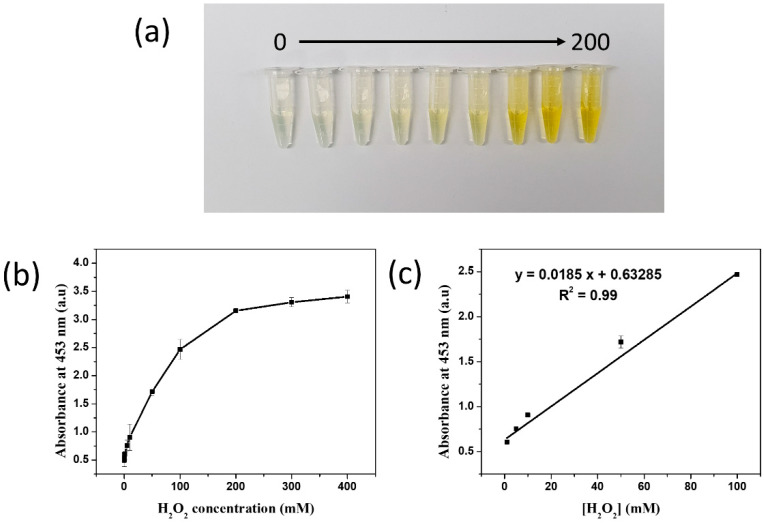
(**a**) Colors of the solutions. (**b**) An absorbance signal plot at 453 nm and (**c**) the dynamic linear range of SiO_2_@Au@Pt@Pt at various H_2_O_2_ concentrations in the presence of 0.5 mM TMB. The optimized conditions were 5 µg SiO_2_@Au@Pt@Pt, 0.5 mM TMB, a 15 min incubation period, and a pH of 4.0.

## Data Availability

Data is available in the manuscript and supporting information.

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
