# Peer review of "Synthesis of Gold-Platinum Core-Shell Nanoparticles Assembled on a Silica Template and Their Peroxidase Nanozyme Properties"

_ijms, 2022, doi:10.3390/ijms23126424_

Round 1
Reviewer 1 Report
Dear Editor,
the manuscript by Xuan-Hung Pham and coworkers deals with the synthesis and catalytic properties of Au@Pd Np supported onto silica NP substrate. It is clearly written and easy to read. The results and conclusions appear consistent.
There are few points that may improve the ...
- In the figure 1b the fluorescence from Au ad Pt are not visible on the dark background. Please change the contrast
- How is the reciprocal distribution of Au and Pt? It would be relevant to assess if Au and Pt coexist or distribute randomly.
- May the author estimate the Su Au@Pt NP sizes from TEM images? It may help understanding the different performances as a function of composition.
- X-ray diffraction, X-ray absorption (XAFS) and/or x-ray photoelectron (XPS) spectroscopy and/or x-ray could provide atomic scale structural and electronic information about the atomic scale mechanisms related to the chemical activities in these catalysts: AuPd alloying, PdO forming, core/shell nature of NP and so on. See as few examples: Nature Communications volume 6, Article number: 6540 (2015); Appl. Sci. 2019, 9(15), 2959; https://doi.org/10.3390/app9152959; Langmuir 2005, 21, 18, 8502–8508. Can the Authors provide further details about the formation of Au-Pt/Pt-O phases, the reciprocal distribution of Au and Pt in the Au@Pt and/or comment about further adequate investigation?
Author Response
We appreciate the comments from the reviewer who spent invaluable time and effort. We have incorporated additional modifications based on the reviewers’ thoughtful comments, which have helped us to improve the manuscript. The detailed responses to the reviewers’ comments are provided at the end of this letter.

Reviewer 2 Report
The Authors described review concerning on synthesis of gold-platinum core-shell nanoparticles assembled on silica template and their peroxidase nanozyme properties. The manuscript could be published in International Journal of Molecular Sciences after major revision. Below, several aspects have mentioned, which should be corrected and some doubts should be explained.
- The Abstract should contain the most important results from the manuscript.
- The motivation of studies should be highlighted in the Introduction.
- What is the physical meaning of fitting in fig. 5?
- The Discussion is extremely poor. The Authors should compare their results to some literature.
Generally, the Authors did an excellent work. However it could not be published in present form in International Journal of Molecular Sciences. I recommend major revision.
Author Response
We appreciate the comments from the reviewer who spent invaluable time and effort. We have incorporated additional modifications based on the reviewers’ thoughtful comments, which have helped us to improve the manuscript. The detailed responses to the reviewers’ comments are provided at the end of this letter.
We appreciate the comments from the reviewer who spent invaluable time and effort. We have incorporated additional modifications based on the reviewers’ thoughtful comments, which have helped us to improve the manuscript. The detailed responses to the reviewers’ comments are provided at the end of this letter.
- The Abstract should contain the most important results from the manuscript.
Thank you for your comment. We rewrite the Abstract as the following.
“Bimetallic nanoparticles are important materials for synthesizing multifunctional nanozymes. A technique for preparing gold-platinum nanoparticles (NPs) on a silica core template (SiO2@Au@Pt) using seed-mediated growth is reported in this study. The SiO2@Au@Pt exhibits peroxidase-like nanozyme activity has several advantages over gold assembled silica core templates (SiO2@Au@Au), such as stability and catalytic performance. The maximum reaction velocity (Vmax) and the Michaelis–Menten constants (Km) were and 2.1 × 10−10 M−1∙s−1 and 417 µM, respectively. Factors affecting the peroxidase activity, including the quantity of NPs, solution pH, reaction time, and concentration of tetramethyl benzidine, are also investigated in this study. The optimization of SiO2@Au@Pt NPs for H2O2 detection obtained in 0.5 mM TMB; using 5 µg SiO2@Au@Pt, at pH 4.0 for 15 mins incubation. H2O2 can be detected in the dynamic liner range of 1.0 to 100 mM with the detection limit of 1.0 mM. This study presents a novel method for controlling the properties of bimetallic NPs assembled on a silica template and increases the understanding of the activity and potential applications of highly efficient multifunctional NP-based nanozymes.”
- The motivation of studies should be highlighted in the Introduction.
Thank you for your comment. We modified the Introduction to make our motivation clear in line 78-84 and line 96-97
“In this study, gold-platinum NPs are prepared on a silica core template (SiO2@Au@Pt) using seed-mediated growth combined with the dropping method and utilize it as a nanomaterial to detect hydrogen peroxide (H2O2) as a model because H2O2 plays an important role in cellular metabolism and in various industry such as food industry, gas sensors, pharmaceuticals, catalysis, environment, and solar energy [59-61]. Therefore, the need to develop a low cost, simple, fast, and sensitive method for monitoring the concentration of H2O2 is of practical significance in both industry and academia.”
“After the optimization of synthesis, SiO2@Au@Pt was used as a peroxidase-like nanomaterial to detect H2O2 efficiently.”
- What is the physical meaning of fitting in fig. 5?
The fitting in the Fig.5 indicates a dynamic linear range of an analytical method. It will facilitate the user to accurately determine the unknown H2O2 concentration in the sample such as cell extract and so on.
- The Discussion is extremely poor. The Authors should compare their results to some literature.
Thank you for your comment. We added some literature (reference 11, 37, 45, 66, 67, 69) to our revised manuscript to explain the increase of catalytic activity of H2O2 in our revised manuscript.
Line 117 -122
“The absorbance intensity of SiO2@Au showed a slight increase at ~500 nm. This peak was suppressed in the absorbance spectrum of SiO2@Au@Pt but the spectrum of SiO2@Au@Pt increased in the UV region (Figure 1c) owing to the Pt layer, consistent with previous results of pure Pt hydrosol [66,67]. The results of TEM, EDS mapping, UV-Vis spectrum of SiO2@Au@Pt indicated that Pt was deposited to the SiO2@Au.”
Line 184 – 195
The size of Pt increased as the Pt2+ concentration increased (Figure 3a and Figure S2). In particularly, the size of Au@Pt synthesized at 100, 200, 300, 400 µM Pt2+ were 3.1 ± 0.61; 3.6 ± 0.56; 3.9 ± 0.59; 4.6 ± 0.62 nm, respectively. At a high concentration of Pt2+ (>200 µM), the Au@Pt on the SiO2 surface partly merged. The quantitative EDS analysis of Pt and Au elements on the surface of SiO2 synthesized at various Pt2+ concentration was carried out to investigate the composition of the Au@Pt on the SiO2 surface and the results were shown in Table S1. All Au@Pt NPs contained both Pt and Au elements, but their Au and Pt components were different. The atomic Pt component increased from 71.59% to 87.07% while the atomic Au component decreased from 28.41 to 12.93% when Pt2+ increased from 100 µM to 400 µM. Therefore, the reciprocal of Pt and Au increased from 2.5 (at 100 µM Pt2+) to 6.7 (at 400 µM Pt2+). The results matched to the TEM images, and it indicated that Pt was gradually deposited on the surface of SiO2@Au.
Line 220 – 232
’Therefore, the catalytic activity of SiO2@Au@Pt NPs increased as the concentration of the Pt2+ precursor increased from 100 to 200 mM because of the formation of sublayers or a monolayer of Pt on the surface of SiO2@Pt [45,67]. This result consistent with the previous report where the increase of Pt component of Au-Pt led an better catalytic activity because the alloying of Pt with Au can change the electronic structure of Pt, leading the catalytic performance of Au@Pt changes [11,37,69]. Although the Au@Pt NPs grew as the concentration of the Pt2+ precursor increased, the peroxidase reaction of SiO2@Au@Pt treated with > 200 µM of the Pt2+ precursor did not increase because TMB and H2O2 cannot gain access to inner part of the thick Pt layer [66]. It means that the catalytic activities of SiO2@Au@Pt reached the highest value at the Pt/Au ratio of 3.2 and decreased when Pt2+ concentration increased further. Therefore, 200 µM of Pt2+ precursor-treated SiO2@Au@Pt NPs, which exhibited high peroxidase-like activity, were used in subsequent experiments.”
We added some literature to compare our detection of limit.
Line 269 -273
This LOD is higher than the LOD of silver NPs modified cellulose nanowhiskers [79], polyoxometalate [80], Ce2(WO4)3, papain [81], Ag-nanoparticle-decorated silica microspheres [82] and magnetic mesoporous silica nanoparticles [83].
Although the LOD is not an advantage of SiO2@Au@Pt but this material is stable and easily handling.
Round 2
Reviewer 2 Report
The Authors improved the manuscript according to comments of reviewers. It could be published in present version.
Author Response
The Authors improved the manuscript according to comments of reviewers. It could be published in present version.
Answer:
We appreciate the reviewer's comments in improving the quality of our manuscript.
Also, we are grateful for your consent to our publication in the Internal Journal of Molecular Sciences.